# CropNet: An Open Large-Scale Dataset with Multiple Modalities for Climate Change-aware Crop Yield Predictions

## Abstract

Precise crop yield predictions are of national importance for ensuring food security and sustainable agricultural practices. While *AI-for-science* approaches have exhibited promising achievements in solving many scientific problems such as drug discovery, precipitation nowcasting, *etc*., the development of deep learning models for predicting crop yields is constantly hindered by the lack of an open and large-scale deep learning-ready dataset with multiple modalities to accommodate sufficient information. To remedy this, we introduce the CropNet dataset, the first *terabyte-sized*, publicly available, and multi-modal dataset specifically targeting crop yield predictions for the contiguous United States (U.S.) continent at the county level. Our CropNet dataset is composed of three modalities of data, *i.e.*, Sentinel-2 Imagery, WRF-HRRR Computed Dataset, and USDA Crop Dataset, for over 2200 U.S. counties spanning 6 years (2017-2022), expected to facilitate researchers in developing versatile deep learning models for timely and precisely predicting crop yields at the county-level, by accounting for the effects of both short-term growing season weather variations and long-term climate change on crop yields. Besides, we release our CropNet package in the Python Package Index (PyPI), with three types of APIs developed for facilitating researchers in downloading the CropNet data on the fly over the time and region of interest, and flexibly building their deep learning models for accurate crop yield predictions. Extensive experiments have been conducted on our CropNet dataset via employing various types of deep learning solutions, with the results validating the general applicability and the efficacy of the CropNet dataset in climate change-aware crop yield predictions. Our dataset is available at https://anonymous.4open.science/r/CropNet, and our CropNet package is available at https://pypi.org/project/cropnet/.

## 1 Introduction

Precise crop yield prediction is essential for early agricultural planning Khaki et al. (2021), timely management policy adjustment Turchetta et al. (2022), informed financial decision making Ansarifar et al. (2021), and national food security Mourtzinis et al. (2021). Inspired by the recent success of deep neural networks (DNNs) Krizhevsky et al. (2012); He et al. (2016); Vaswani et al. (2017); Dosovitskiy et al. (2021), plenty of studies Khaki et al. (2020; 2021); Garnot & Landrieu (2021); Wu et al. (2021); Cheng et al. (2022); Fan et al. (2022); Lin et al. (2023) have adopted deep learning (DL) models for timely and precisely predicting crop yields. However, they often applied their personally curated and limit-sized datasets, with somewhat mediocre prediction performance. There is an urgent need for new large-scale and deep learning-ready datasets tailored specifically for wide use in crop yield predictions.

Recently, some studies Veillette et al. (2020); Garnot & Landrieu (2021); Tseng et al. (2021); Cornebise et al. (2022); Chen et al. (2022); Ashkboos et al. (2022) have developed open and large-scale satellite imagery (or meteorological parameter) datasets, flexible for being adopted to agricultural-related tasks, *e.g*., crop type classification Tseng et al. (2021). Unfortunately, two limitations impede us from applying them directly to crop yield predictions in general. First, they lack ground-truth crop yield information, making them unsuitable for crop yield predictions. Second,

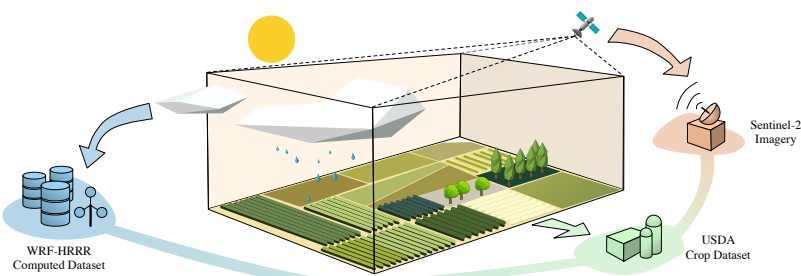

Figure 1: Our CropNet dataset is composed of three modalities of data, *i.e.*, Sentinel-2 Imagery, WRF-HRRR Computed Dataset, and USDA Crop Dataset, providing satellite images, meteorological parameters, and county-level crop yield information, respectively.

they provide only one modality of data (*i.e.*, either satellite images or meteorological parameters), while accurate crop yield predictions often need to track the crop growth and capture the meteorological weather variation effects on crop yields simultaneously, calling for multiple modalities of data. To date, the development of a large-scale dataset with multiple modalities, targeting specifically for county-level crop yield predictions remains open and challenging.

In this work, we aim to craft such a dataset, called CropNet, the first *terabyte-sized* and publicly available dataset with multiple modalities, designed specifically for county-level crop yield predictions across the United States (U.S.) continent. As shown in Figure 1, the CropNet dataset is composed of three modalities of data, *i.e.*, Sentinel-2 Imagery, WRF-HRRR Computed Dataset, and USDA Crop Dataset, covering a total of 2291 U.S. counties from 2017 to 2022[1]. In particular, the Sentinel-2

Table 1: Dataset comparison

| Dataset | Size (GB) | Data Modality |
|---|---|---|
| SEVIR Veillette et al. (2020) | 970 | satellite imagery |
| DENETHOR Kondmann et al. (2021) | 254 | satellite imagery |
| PASTIS Garnot & Landrieu (2021) | 29 | satellite imagery |
| WorldStrat Cornebise et al. (2022) | 107 | satellite imagery |
| RainNet Chen et al. (2022) | 360 | satellite imagery |
| ENS-10 Ashkboos et al. (2022) | 3072 | meteorological parameters |
| Our CropNet dataset | 2362 | satellite imagery meteorological parameters crop information |

Imagery, acquired from the Sentinel-2 mission Sentinel-2 (2023), provides two categories of satellite images, *i.e.*, agriculture imagery (AG) and normalized difference vegetation index (NDVI), for precisely monitoring the crop growth on the ground. The WRF-HRRR Computed Dataset, obtained from the WRF-HRRR model HRRR (2023), offers daily and monthly meteorological parameters, accounting respectively for the short-term weather variations and the long-term climate change. The USDA Crop Dataset, sourced from the USDA Quick Statistic website USDA (2023), contains annual crop yield information for four major crops, *i.e.*, corn, cotton, soybean, and winter wheat, grown on the contiguous U.S. continent, serving as the ground-truth label for crop yield prediction tasks. Table 1 summarizes the dataset comparison between our CropNet dataset and pertinent datasets.

Since the data in our CropNet dataset are obtained from different data sources, we propose a novel data alignment solution to make Sentinel-2 Imagery, WRF-HRRR data, and USDA crop yield data spatially and temporally aligned. Meanwhile, three modalities of data are stored in carefully designed file formats, for improving the accessibility, readability, and storage efficiency of our CropNet dataset. The key advantage of our CropNet dataset is to facilitate researchers in developing crop yield prediction models that are aware of climate change, by taking into account the effects of (1) the short-term weather variations, governed by daily parameters during the growing season, and (2) the long-term climate change, governed by monthly historical weather variations, on crop growth. Furthermore, we release our CropNet package, including three types of APIs, in the Python Package Index (PyPI), enabling researchers and practitioners to (1) dynamically download the CropNet data based on the specific time and region of interest and (2) flexibly develop climate change-aware deep learning models for accurate crop yield predictions at the county level.

Our experimental results validate that the CropNet dataset can be easily adopted by the prominent deep learning models, such as Long Short-Term Memory (LSTM)-based, Convolutional Neural Network (CNN)-based, Graph Neural Network Kipf & Welling (2017) (GNN)-based, and Vision Transformer Dosovitskiy et al. (2021) (ViT)-based models, for timely and precise crop yield predictions. Additionally, our CropNet dataset demonstrates its versatile applicability to both unimodal (*i.e.*, vi-

---

[1]Our dataset is currently up-to-date as the majority of USDA crop information for 2023 is not yet available.

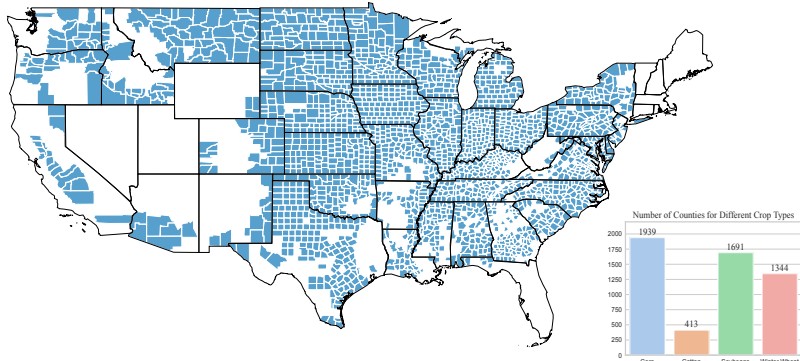

Figure 2: Geographic distribution of our CropNet dataset across 2291 U.S. counties. The rightmost bar chart provides the number of counties corresponding to each of the four crop types included in the USDA Crop Dataset.

Table 2: Overview of our CropNet dataset

| Data Modality | Size | Spatial Resolution | Temporal Resolution | Content |
|---|---|---|---|---|
| Sentinel-2 Imagery | 2326.7 GB | 9x9 km | 14 days | 224x224 RGB satellite images |
| WRF-HRRR Computed Dataset | 35.5 GB | 9x9 km | 1 day or 1 month | 9 weather parameters |
| USDA Crop Dataset | 2.3 MB | county-level | 1 year | 2 crop parameters |

sual) and multi-modal (*i.e.*, visual and numerical) self-supervised pre-training scenarios, thanks to its abundant visual satellite imagery and numerical meteorological data.

## 2    OUR CROPNET DATESET

Our CropNet dataset is crafted from three different data sources, *i.e.*, the Sentinel-2 mission Sentinel-2 (2023), the Weather Research & Forecasting-based High-Resolution Rapid Refresh Model (WRF-HRRR Model) HRRR (2023), and the United States Department of Agriculture (USDA) USDA (2023), with their details deferred to Appendix A.

### 2.1    MOTIVATION

The large-scale data with multiple modalities comprising satellite images, numerical meteorological weather data, and crop yield statistic data, are essential for tracking crop growth and correlating the weather variation's effects on crop yields, to be used for timely and precisely predicting crop yields at the county level. To date, such an open and multi-modal dataset intended for county-level crop yield prediction is still absent. In this benchmark article, we plan to design and publish such an open and large-scale dataset, called CropNet, with multiple modalities, consisting of visual satellite images, numerical meteorological parameters, and crop yield statistic data, across the U.S. continent. Notably, not all U.S. counties are suitable for crop planting, so our dataset only includes the data corresponding to 2291 U.S. counties over 3143 counties in total (see Figure 2 for its geographic distribution). Such a multi-modal dataset is valuable for researchers and practitioners to design and test various deep learning models for crop yield predictions, by taking into account the effects of both short-term growing season weather variations and long-term climate change on crop yields.

### 2.2    OVERVIEW OF OUR CROPNET DATASET

Our CropNet dataset is composed of three modalities of data, *i.e.*, Sentinel-2 Imagery, WRF-HRRR Computed Dataset, and USDA Crop Dataset, spanning from 2017 to 2022 (*i.e.*, 6 years) across 2291 U.S. counties. Figure 2 shows the geographic distribution of our dataset. Since crop planting is highly geography-dependent, Figure 2 also provides the number of counties corresponding to each crop type in the USDA Crop Dataset (see the rightmost bar chart). Notably, four of the most popular crops, *i.e.*, corn, cotton, soybeans, and winter wheat, are included in our CropNet dataset, with satellite imagery and the meteorological data covering all 2291 counties. Table 2 overviews our CropNet dataset. Its total size is 2362.6 GB, with 2326.7 GB of visual data for Sentinel-2 Imagery,

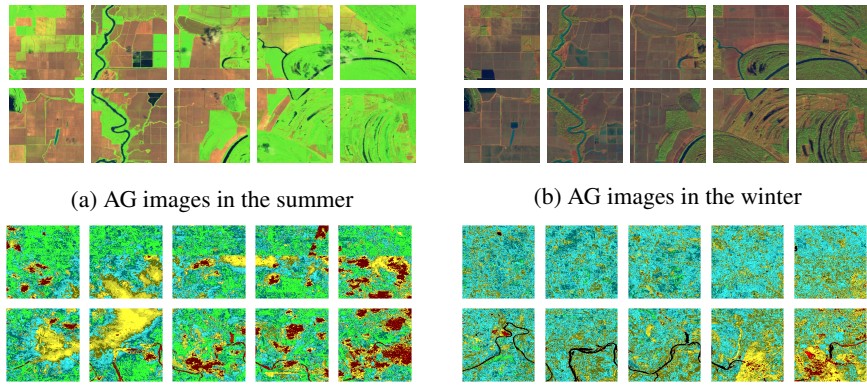

(a) AG images in the summer      (b) AG images in the winter

(c) NDVI images in the summer      (d) NDVI images in the winter

Figure 3: Examples of agriculture imagery (AG, see 3a and 3b) and normalized difference vegetation index (NDVI, see 3c and 3d) in Sentinel-2 Imagery.

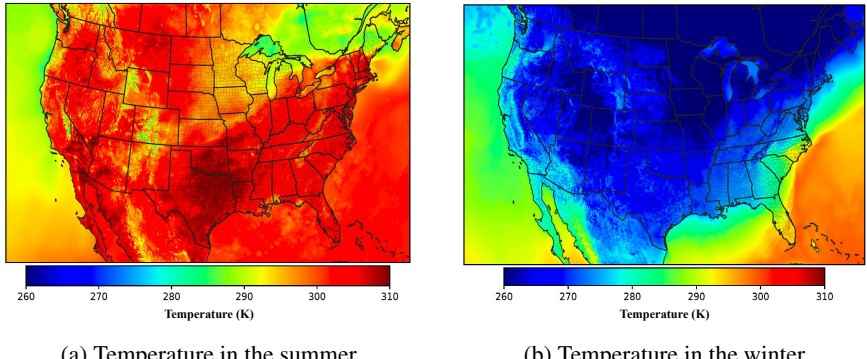

(a) Temperature in the summer      (b) Temperature in the winter

Figure 4: Examples of the temperature parameters in the WRF-HRRR Computed Dataset.

**USDA Crop Dataset: 2022 Soybeans Yield**

Figure 5: Illustration of USDA Crop Dataset for 2022 soybeans yields.

35.5 GB of numerical data for WRF-HRRR Computed Dataset, and 2.3 MB of numerical data for USDA Crop Dataset. Specifically, Sentinel-2 Imagery contains two types of satellite images (*i.e.*, AG and NDVI), both with a resolution of 224x224 pixels, a spatial resolution of 9x9 km, and a revisit frequency of 14 days. Figures 3a (or 3b) and 3c (or 3d) respectively depict examples of AG and NDVI images in the summer (or winter). The WRF-HRRR Computed Dataset provides daily (or monthly) meteorological parameters gridded at the spatial resolution of 9 km in a one-day (or one-month) interval. Figures 4a and 4b visualize the temperature in the WRF-HRRR Computed Dataset for the summer and the winter, respectively. The USDA Dataset offers crop information for four types of crops each on the county-level basis, with a temporal resolution of one year. Figure 5 shows the example for the USDA Crop Dataset, depicting 2022 soybeans yields across the U.S. continent. More examples of our CropNet dataset are deferred to Appendix B.

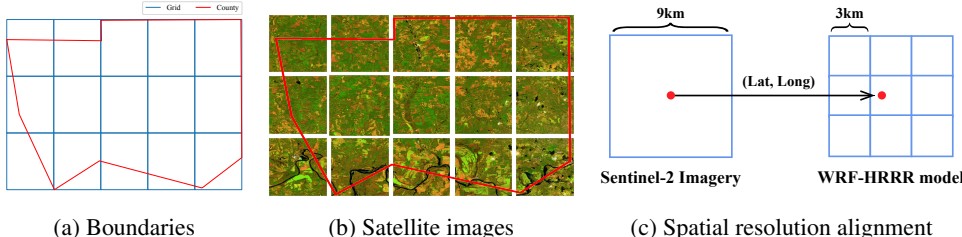

(a) Boundaries        (b) Satellite images        (c) Spatial resolution alignment

Figure 6: Illustration of county partitioning (*i.e.*, 6a and 6b) and spatial resolution alignment (*i.e.*, 6c). (a) Boundaries for one county (*i.e.*, the red line) and the corresponding high-resolution grids (*i.e.*, the blue line). (b) Satellite images in the Sentinel-2 Imagery for representing the county. (c) One 3x3km and its surrounding eight grids in the WRF-HRRR model are used for aligning with one 9x9km grid in the Sentinel-2 Imagery.

## 2.3 DATA COLLECTION AND PREPARATION

**Sentinel-2 Imagery.** We utilize the Sentinel Hub Processing API Sentinel-Hub (2023) to acquire satellite images from the Sentinel-2 mission at a processing level of Sentinel-2 L1C, with a maximum allowable cloud coverage of 20%, three spectral bands (*i.e.*, B02, B08, and B11) for AG images and two bands (*i.e.*, B04 and B08) for NDVI images. Satellite images are obtained at the revisit frequency of 14 days instead of the original highest revisit frequency of 5 days. The reason is that the 5-day revisit frequency under our cloud coverage setting results in a large number of duplicate satellite images, according to our empirical study (refer to Appendix C.1 for details). As precisely tracking the crop growth on the ground requires high-spatial-resolution satellite images, we partition a county into multiple grids at the spatial resolution of 9x9 km, with each grid corresponding to one satellite image. Figures 6a and 6b illustrates an example of county partitioning (refer to Appendix C.2 for more details). The downloaded satellite images for one U.S. state (including all counties therein) spanning one season are stored in one Hierarchical Data Format (HDF5) file. Three reasons motivate us to employ the HDF5 file format. First, it can significantly save the hard disk space. That is, the collected satellite images with a total of 4562.2 GB shrank to 2326.7 GB (*i.e.*, 0.51x smaller space occupancy) in the HDF5 file. This can facilitate researchers and practitioners for lower hard disk space requirements and faster data retrieval. Second, it allows for storing data in the form of multidimensional arrays, making satellite images easy to access. The HDF5 file for Sentinel-2 Imagery is organized in the form of $(F, T, G, H, W, C)$, where $F$ represents the FIPS code (*i.e.*, the unique number for each U.S. county) used for retrieving one county's data, $T$ indicates the number of temporal data in a 14-day interval with respect to one season, $G$ represents the number of high-resolution grids for a county, and $(H, W, C)$ are the width, height, and channel numbers for the satellite image. Third, it can store descriptive information for the satellite image, such as its revisit day, the latitude and longitude information it represents, among others.

**WRF-HRRR Computed Dataset.** The WRF-HRRR Computed Dataset is sourced from the WRF-HRRR model HRRR (2023), which produces GRID files on the hourly basis, containing meteorological parameters over the contiguous U.S. continent at a spatial resolution of 3x3 km. To lift the domain knowledge required for using the WRF-HRRR data, our CropNet dataset includes 9 carefully chosen and crop growth-relevant meteorological parameters, with 6 parameters obtained directly from the WRF-HRRR model, *i.e.*, averaged temperature, precipitation, relative humidity, wind gust, wind speed, downward shortwave radiation flux, and other 3 parameters computed by ourselves, *i.e.*, maximal temperature, minimal temperature, vapor pressure deficit (VPD). More details of the WRF-HRRR Computed Dataset, including equations for calculating VPD, are deferred to Appendix C.3.

Two challenges impede us from efficiently and effectively extracting meteorological parameters from GRID files. First, the spatial resolution in the WRF-HRRR Computed Dataset should align with the one in the Sentinel-2 Imagery, *i.e.*, 9x9 km$^2$. A novel solution is proposed to address this issue. We first follow the Sentinel-2 Imagery by partitioning one county into multiple grids at the spatial resolution of 9x9 km. Then, we utilize the latitude and longitude of the centric point in the 9x9km grid to find the nearest 3x3km grid in the WRF-HRRR model. Next, meteorological

---

[2]Note that acquiring satellite images at a spatial resolution of 3x3km is infeasible in practice due to its tremendous space size requirement (*i.e.*, over 20 TB).

parameters in the 3x3 km grid and its surrounding 8 grids can be used for representing a region gridded at 9x9 km, as shown in Figure 6c. In this way, our dataset allows researchers to capture the immediate effects of atmospheric weather variations occurring directly above the crop-growing area on crop yields. Second, extracting meteorological parameters from GRID files is extremely time-consuming as searching the nearest grids requires to match geo-grids across the continental United States. To handle this challenge, we develop a global cache solution by pre-storing the nearest grid information corresponding to a pair of latitude and longitude for each location, reducing the required extraction time from 60 days to 42 days (*i.e.*, 1.42x faster than the one without global caching).

The daily meteorological parameters are computed out of the hourly data extracted from the GRID file, while the monthly weather parameters are derived from our daily data to significantly reduce the frequency of accessing the GRID file. Finally, daily and monthly meteorological parameters are stored in the Comma Separated Values (CSV) file, making them readable by researchers and accessible for deep learning models. The CSV file also includes additional valuable information such as the FIPS code of a county and the latitude and longitude of each grid. This provides easy and convenient accesses to relevant data for researchers.

**USDA Crop Dataset.** The data in the USDA Crop Dataset is retrieved from the USDA Quick Statistic website USDA (2023) via our newly developed web crawler solution. For each crop type, the USDA website provides its crop information at the county level in a one-year interval, with a unique key for identifying the data for one crop type per year, *e.g.*, "85BEE64A-E605-3509-B60C-5836F6FBB5F6" for the corn data in 2022. Our web crawler first retrieves the unique key by specifying the crop type and the year we need. Then, it utilizes the unique key to obtain the corresponding crop data in one year. Finally, the downloaded crop data is stored in the CSV file. Notably, other useful descriptive information, *e.g.*, FIPS code, state name, county name, *etc.*, are also contained in the CSV file for facilitating readability and accessibility.

However, the crop statistic data from the USDA Quick Statistic website is not deep learning-friendly. For example, it uses two columns, *i.e.*, "Data Item" and "Value", to keep all valuable crop information. That is, if the description of the "Data Item" column refers to the corn yield, then the numerical data in the "Value" column represents the corn yield. Otherwise, the data in "Value" may signify other information, *e.g.*, the corn production, the soybeans yield, *etc*. New data pre-processing techniques are developed to unify the data format, making the production and yield information stored in two independent columns for facilitating Python libraries (*e.g.*, pandas) to access them.

Our CropNet dataset specifically targets county-level crop yield predictions across the contiguous U.S. continent. We utilize the FIPS code to rapidly fetch the data of each county, including a list of HDF5 files for Sentinel-2 Imagery, two lists of CVS files respectively for daily and monthly meteorological parameters, and one CVS file for the USDA Crop Dataset, with configurations stored in the JSON file for increasing accessibility (see Appendix C.4 for an example of our JSON configuration file).

## 3 EXPERIMENTS AND RESULTS

Three scenarios of climate change-aware crop yield predictions, *i.e.*, **Crop Yield Predictions**, **One-Year Ahead Predictions**, and **Self-Supervised Pre-training**, are considered to exhibit the general applicability of our CropNet dataset to various types of deep learning solutions.

### 3.1 EXPERIMENTAL SETTINGS

**Approaches.** The LSTM-based, CNN-based, GNN-based, and ViT-based models are represented respectively by **ConvLSTM** Shi et al. (2015), **CNN-RNN** Khaki et al. (2020), **GNN-RNN** Fan et al. (2022), and **MMST-ViT** Lin et al. (2023) in our experiments, targeting crop yield predictions. Meanwhile, two self-supervised learning (SSL) techniques, *i.e.*, **MAE** He et al. (2022), and **MM-SSL** in the MMST-ViT, serving respectively as unimodal and multi-modal SSL techniques, are taken into account under the self-supervised pre-training scenario. The aforementioned methods are modified slightly to make them fit the CropNet data in our experiments.

**Metrics.** Three performance metrics, *i.e.*, **Root Mean Square Error (RMSE)**, **R-squared ($R^2$)**, and **Pearson Correlation Coefficient (Corr)**, are adopted to evaluate the efficacy of the CropNet

Table 3: Overall performance for 2022 crop yield predictions, where the yield of cotton is measured in pounds per acre (LB/AC) and those of the rest are measured in bushels per acre (BU/AC)

| Method | Corn | | | Cotton | | | Soybeans | | | Winter Wheat | | |
|---|---|---|---|---|---|---|---|---|---|---|---|---|
| | RMSE ($\downarrow$) | $R^2$ ($\uparrow$) | Corr ($\uparrow$) | RMSE ($\downarrow$) | $R^2$ ($\uparrow$) | Corr ($\uparrow$) | RMSE ($\downarrow$) | $R^2$ ($\uparrow$) | Corr ($\uparrow$) | RMSE ($\downarrow$) | $R^2$ ($\uparrow$) | Corr ($\uparrow$) |
| ConvLSTM | 19.2 | 0.795 | 0.892 | 56.7 | 0.834 | 0.913 | 5.3 | 0.801 | 0.895 | 6.0 | 0.798 | 0.893 |
| CNN-RNN | 14.3 | 0.867 | 0.923 | 54.5 | 0.826 | 0.899 | 4.1 | 0.853 | 0.915 | 5.6 | 0.823 | 0.906 |
| GNN-RNN | 14.1 | 0.871 | 0.917 | 55.1 | 0.813 | 0.881 | 4.1 | 0.868 | 0.929 | 5.3 | 0.845 | 0.912 |
| MMST-ViT | 13.2 | 0.890 | 0.943 | 50.9 | 0.848 | 0.921 | 3.9 | 0.879 | 0.937 | 4.8 | 0.864 | 0.929 |

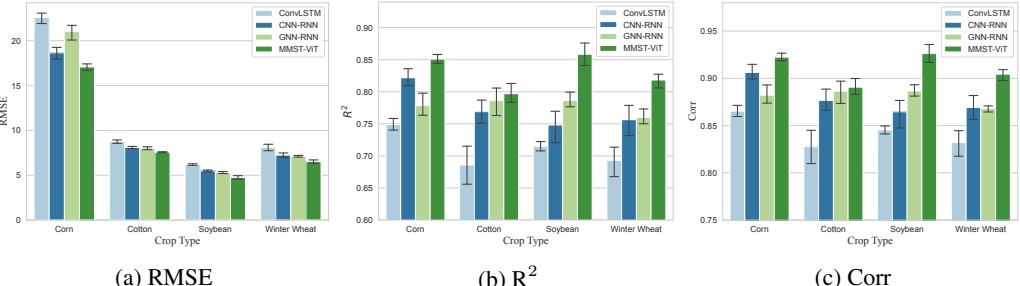

(a) RMSE      (b) $R^2$      (c) Corr

Figure 7: The performance of one-year ahead crop yield predictions, with the cotton yield measured by LB/AC and other crop yields measured by BU/AC. In Figure 7a, we present the square root of the RMSE values for the cotton yield to enhance visualization.

dataset for crop yield predictions. Note that a lower RMSE value and a higher $R^2$ (or Corr) value represent better prediction performance.

Details of utilizing our CropNet data for conducting experiments is deferred to Appendix D.1.

## 3.2 PERFORMANCE EVALUATION FOR 2022 CROP YIELD PREDICTIONS

We conduct experiments on the CropNet dataset for 2022 crop yield predictions by using satellite images and daily weather conditions during growing seasons, as well as monthly meteorological conditions from 2017 to 2021, running under the ConvLSTM, CNN-RNN, GNN-RNN, and MMST-ViT models. Table 3 presents each crop's overall performance results (*i.e.*, RMSE, $R^2$, and Corr) in aggregation. We have two observations. First, all models achieve superb prediction performance with our CropNet data. For example, ConvLSTM, CNN-RNN, GNN-RNN, and MMST-ViT achieve small RMSE values of 5.3, 4.1, 4.1, and 3.9, respectively, for soybeans yield predictions (see the 8th column). These results validate that our CropNet dataset is well-suited for LSTM-based, CNN-based, and GNN-based, and ViT-based models, demonstrating its general applicability. Second, MMST-ViT achieves the best performance results under all scenarios, with lowest RMSE values of 13.2, 50.9, 3.9, and 4.8, as well as highest $R^2$ (or Corr) values of 0.890 (or 0.943), 0.848 (or 0.921), 0.879 (or 0.937), and 0.864 (or 0.929), respectively for predicting corn, cotton, soybeans, and winter wheat yields. This is due to MMST-ViT's novel attention mechanisms Vaswani et al. (2017), which perform the cross-attention between satellite images and meteorological parameters, able to capture the effects of both growing season weather variations and climate change on crop growth. This experiment exhibits that our CropNet dataset can provide crop yield predictions timely and precisely, essential for making informed economic decisions, optimizing agricultural resource allocation, *etc*.

## 3.3 PERFORMANCE OF ONE-YEAR AHEAD PREDICTIONS

Crop yield predictions well in advance of the planting season are also critical for farmers to make early crop planting and management plans. Here, we apply the CropNet dataset one year before the planting season for predicting the next year's crop yields. Figure 7 shows our experimental results for 2022 crop yield predictions by using our CropNet data during the 2021 growing season. We observe that all models can still maintain decent prediction performance. For instance, ConvLSTM, CNN-RNN, GNN-RNN, and MMST-ViT achieve the averaged RMSE values of 6.2, of 5.4, of 5.3, and of 4.7, respectively, for soybeans predictions. Meanwhile, MMST-ViT consistently achieves excellent Corr values, averaging at 0.922 for corn, 0.890 for cotton, 0.926 for soybeans, and 0.904 for winter wheat predictions, only slightly inferior to the performance results for the regular 2022 crop yield predictions (see the last row in Table 3). This can be attributed to MMST-ViT's ability

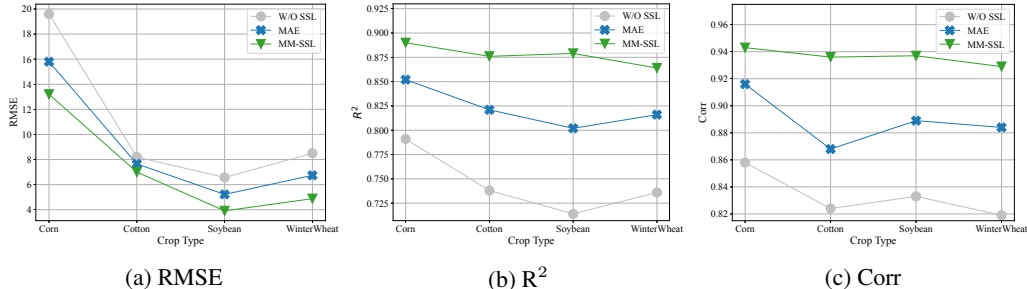

| (a) RMSE | (b) $R^2$ | (c) Corr |

Figure 8: Illustration of how our CropNet dataset benefits self-supervised learning techniques. Notably, Figure 8a depict the square root of RMSE values for the cotton yield to improve visualization.

to capture the indirect influence of 2021's weather conditions on crop growth in the subsequent year through the utilization of long-term weather parameters, which further underscores how our CropNet dataset enhances climate change-aware crop yield predictions.

### 3.4 APPLICATIONS TO SELF-SUPERVISED PRE-TRAINING

Self-supervised learning (SSL) techniques Chen et al. (2020); Bao et al. (2022); He et al. (2022) have shown great promise in addressing the overfitting issue of deep neural networks (DNNs), particularly for vision transformers (ViT) Dosovitskiy et al. (2021). Our CropNet dataset with a total size of over 2 TB of data can benefit both deep-learning and agricultural communities by providing large-scale visual satellite imagery and numerical meteorological data for pre-training DNNs. To exhibit the applications of our CropNet dataset to self-supervised pre-training, we adopt the MMST-ViT for crop yield predictions by considering three scenarios, *i.e.*, MMST-ViT without the SSL technique (denoted as "w/o SSL"), MMST-ViT with the SSL technique in MAE (denoted as "MAE"), and MMST-ViT with the multi-modal SSL technique proposed in Lin et al. (2023) (denoted as "MM-SSL"). Figure 8 illustrates the performance results for four crop types under three performance metrics of interest (*i.e.*, RMSE, $R^2$, and Corr). We discover that without the SSL technique (*i.e.*, the gray line), MMST-ViT suffers from the overfitting issue, achieving the worst crop yield prediction performance under all scenarios. Besides, pre-training MMST-ViT with the SSL technique in MAE (*i.e.*, the blue line) improves its performance results (compared to the "w/o SSL"), with decreased RMSE values by 3.8, 9.6, 1.3, and 1.7 for corn, cotton, soybeans, and winter wheat predictions, respectively. This statistical evidence confirms that our CropNet dataset can be applied to visual SSL techniques. Furthermore, MMST-ViT with the multi-modal SSL technique (*i.e.*, the green line) achieves the best performance results under all scenarios. In comparison to the "w/o SSL" scenario, it decreases RMSE values by 6.4, 18.3, 2.6, and 3.6, respectively, for predicting corn, cotton, soybeans, and winter wheat. This is because the multi-modal SSL technique can leverage both visual satellite images and numerical meteorological data in the CropNet dataset, empowering MMST-ViT to capture the impact of weather conditions on crop growth during the pre-training stage. These results conclusively demonstrate the versatile utility of our CropNet dataset in both unimodal (*i.e.*, visual) and multi-modal (*i.e.*, visual and numerical) SSL pre-training scenarios.

More experimental results, including significance of each modality and impact of long-term meteorological data, are deferred to Appendix D.

## 4 THE CROPNET PACKAGE

In addition to our CropNet dataset, we also release the *CropNet* package, including three types of APIs, at the Python Package Index (PyPI), which is designed to facilitate researchers in developing DNNs for climate change-aware crop yield predictions, with its details presented as follows.

**DataDownloader.** This API allows researchers to download the CropNet data over the time/region of interest on the fly. For example, given the time and region (*e.g.*, the FIPS code for one U.S. county) of interest, Listing 1 presents how to utilize the DataDownloader API to download the up-to-date CropNet data.

**DataRetriever.** This API enables researchers to conveniently obtain the CropNet data stored in the local machine (*e.g.*, after you have downloaded our curated CropNet dataset) over the time/region of interest, with the requested data presented in a user-friendly format. For instance, Listing 2 shows how to employ the DataRetriever API to obtain the CropNet data for two U.S. counties.

**DataLoader.** This API is designed to assist researchers in their development of DNNs for crop yield predictions. It allows researchers to flexibly and seamlessly merge multiple modalities of CropNet data, and then expose them through a DataLoader object after performing necessary data preprocessing techniques. A PyTorch example of using our DataLoader API for training (or testing) DNNs is shown in Listing 3.

```
# Download the Sentinel-2 Imagery for the first three quarters of 2023
downloader.download_Sentinel2(fips_codes=["01003"], years=["2023"])

# Download the WRF-HRRR Computed data for 2023 (January to September)
downloader.download_HRRR(fips_codes=["01003"], years=["2023"])

# Download the 2022 USDA Soybean data
# Note that the majority of 2023 USDA data is currently unavailable
downloader.download_USDA("Soybean", fips_codes=["01003"], years=["2022"])
```

Listing 1: Example of our DataDownloader API.

```
# Retrieve the 2022 USDA data for two U.S. counties
retriever.retrieve_USDA(fips_codes=["01003", "01005"], years=["2022"])

# Retrieve the 2022 Sentinel-2 Imagery data for two U.S. counties
retriever.retrieve_Sentinel2(fips_codes=["01003","01005"],years=["2022"])

# Retrieve the 2022 WRF-HRRR Computed data for two U.S. counties
retriever.retrieve_HRRR(fips_codes=["01003", "01005"], years=["2022"])
```

Listing 2: Example of our DataRetriever API.

```
from torch.utils.data import DataLoader

# The base directory for the CropNet dataset
base_dir = "/mnt/data/CropNet"
# The JSON configuration file
config_file = "data/soybeans_train.json"

# The PyTorch dataloaders for each modality of data
sentinel2_loader = DataLoader(Sentinel2Imagery(base_dir, config_file))
hrrr_loader = DataLoader(HRRRComputedDataset(base_dir, config_file))
usda_loader = DataLoader(USDACropDataset(base_dir, config_file))
```

Listing 3: The PyTorch example of our DataLoader API.

## 5 CONCLUSION

This work presented our crafted CropNet dataset, an open, large-scale, and multi-modal dataset targeting specifically at county-level crop yield predictions across the contiguous United States continent. Our CropNet dataset is composed of three modalities of data, *i.e.*, Sentinel-2 Imagery, WRF-HRRR Computed Dataset, and USDA Crop Dataset, containing high-resolution satellite images, daily and monthly meteorological conditions, and crop yield information, aligned in both the spatial and the temporal domains. Such a dataset is ready for wide use in deep learning, agriculture, and meteorology areas, for developing new solutions and models for crop yield predictions, with the consideration of both the effects of growing season weather variations and climate change on crop growth. Extensive experimental results validate the general applicability of our CropNet dataset to various types of deep learning models for both the timely and one-year ahead crop yield predictions. Besides, the applications of our CropNet dataset to self-supervised pre-training scenarios demonstrate the datset's versatile utility in both unimodal and multi-modal self-supervised learning techniques. In addition to our crafted dataset, we have also made our CropNet package available on the Python Package Index (PyPI). This package allows researchers and practitioners to (1) construct the CropNet data on the fly over the time/region of interest and (2) flexibly build their deep learning models for climate change-aware crop yield predictions. Although our initial goal of crafting the CropNet dataset and developing the CropNet package is for precise crop yield prediction, we believe its future applicability is broad and deserved further exploration. It can benefit the deep learning, agriculture, and meteorology communities, in the pursuit of more interesting, critical, and pertinent applications.

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

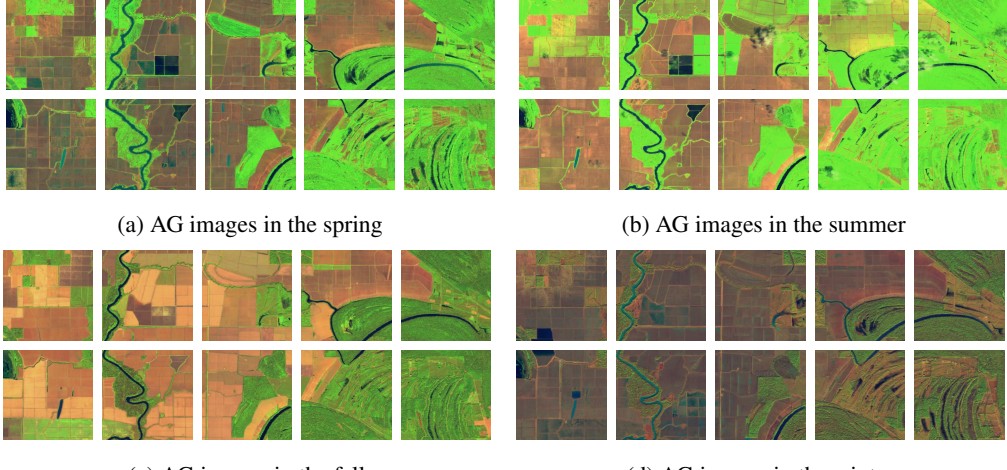

(a) AG images in the spring     (b) AG images in the summer

(c) AG images in the fall     (d) AG images in the winter

Figure 9: Examples of agricultural imagery (AG) from Sentinel-2 Imagery.

## OUTLINE

This document provided supplementary materials to support our main paper. In particular, Section A offers the details of three data sources. Section B illustrates more examples of Our CropNet dataset. Section C provides details of data collection. Section D presents additional experimental settings and results.

## A   DATA SOURCES

Our CropNet dataset is crafted from three different data sources, as listed below.

**Sentinel-2 Mission.** The Sentinel-2 mission Sentinel-2 (2023), launched in 2015, serves as an essential earth observation endeavor. With its 13 spectral bands and high revisit frequency of 5 days, the Sentinel-2 mission provides wide-swath, high-resolution, multi-spectral satellite images for a wide range of applications, such as climate change, agricultural monitoring, *etc*.

**WRF-HRRR Model.** The High-Resolution Rapid Refresh (HRRR) HRRR (2023) is a Weather Research & Forecasting Model (WRF)-based forecast modeling system, which hourly forecasts weather parameters for the whole United States continent with a spatial resolution of 3km. We take the HRRR assimilated results starting from July 2016 and archived in the University of Utah for use, which provides several crop growth-related parameters, *e.g.*, temperature, precipitation, wind speed, relative humidity, radiation, *etc*.

**USDA.** The United States Department of Agriculture (USDA) USDA (2023) provides annual crop information for major crops grown in the U.S., including corn, cotton, soybeans, wheat, *etc.*, at the county level. The statistical data include the planted areas, the harvested areas, the production, and the yield for each type of crop, dated back to 1850 at the earliest.

## B   EXAMPLES OF OUR CROPNET DATASET

This section supplements Section 3.3 of our main paper by providing more examples regarding three modalities of data in our CropNet dataset. First, Figures 9 and 10 respectively illustrate Agricultural (AG) and Normalized Difference Vegetation Index (NDVI) images from Sentinel-2 Imagery under different seasons. Second, Figure 11 shows examples from the HRRR Computed Dataset, with the temperature in Spring, Summer, Fall, and Winter depicted respectively in Figures 11a, 11b, 11c, and 11d. Third, Figure 12 provides 2022 crop yield information from the USDA Crop Dataset, with the corn, cotton, soybeans, and winter wheat yields shown in Figures 12a, 12b, 12c, and 12d, respectively. Note that Our CropNet Dataset also includes crop production data for corn, cotton,

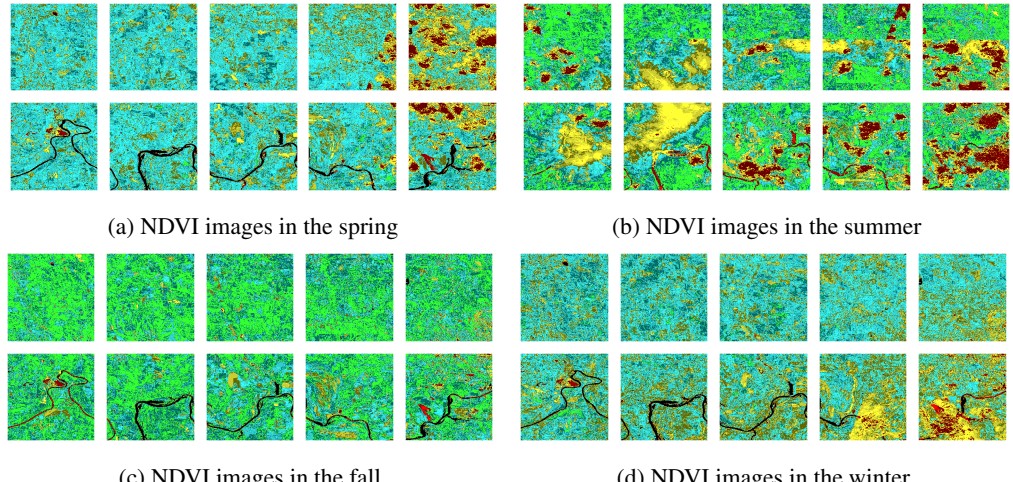

(a) NDVI images in the spring

(b) NDVI images in the summer

(c) NDVI images in the fall

(d) NDVI images in the winter

Figure 10: Examples of normalized difference vegetation index (NDVI) from Sentinel-2 Imagery.

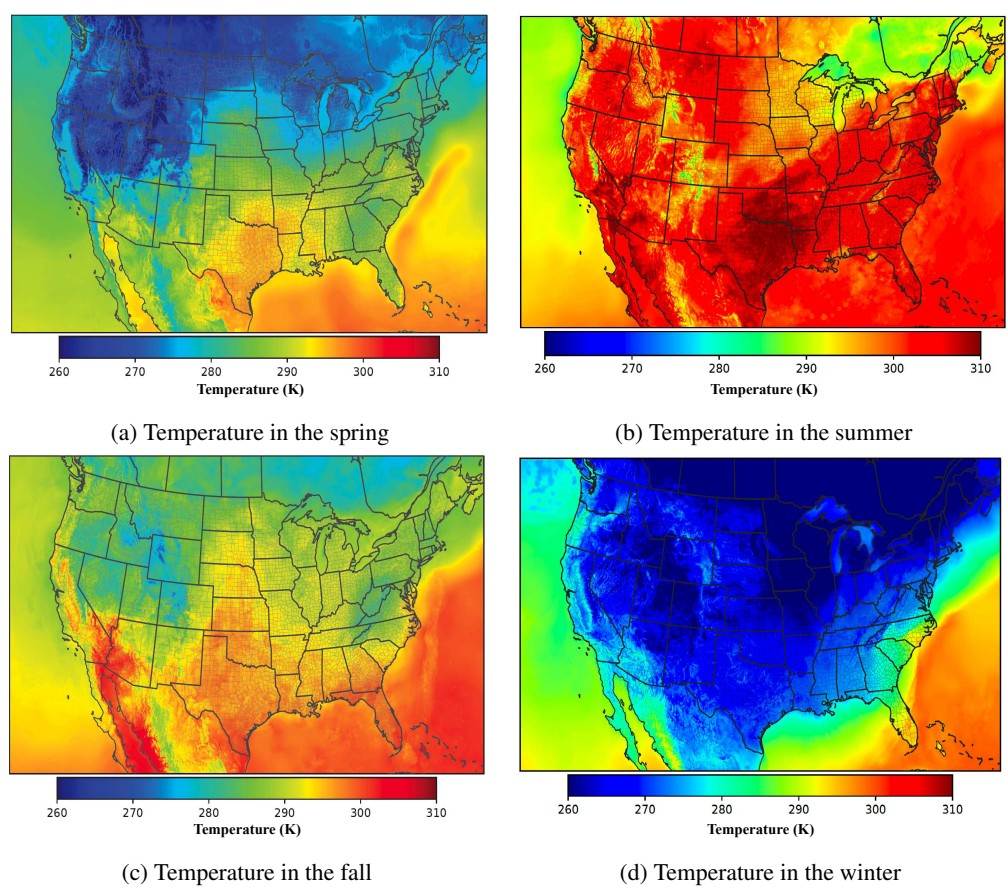

(a) Temperature in the spring

(b) Temperature in the summer

(c) Temperature in the fall

(d) Temperature in the winter

Figure 11: Illustration of the WRF-HRRR Computed Dataset for temperature in (a) the spring, (b) the summer, (c) the fall, and (d) the winter.

soybeans, and winter wheat, with examples regarding their 2022 production information illustrated respectively in Figures 13a, 13b, 13c and 13d.

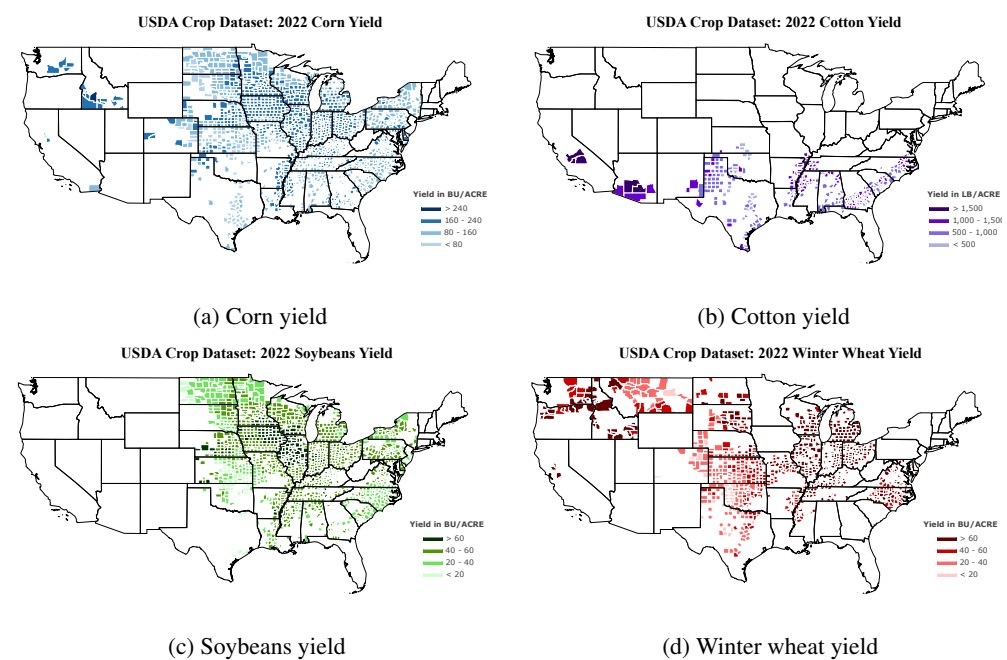

Figure 12: Illustration of USDA Crop Dataset for (a) 2022 corn yield, (b) 2022 cotton yield, (c) 2022 soybeans yield, and (d) 2022 winter wheat yield.

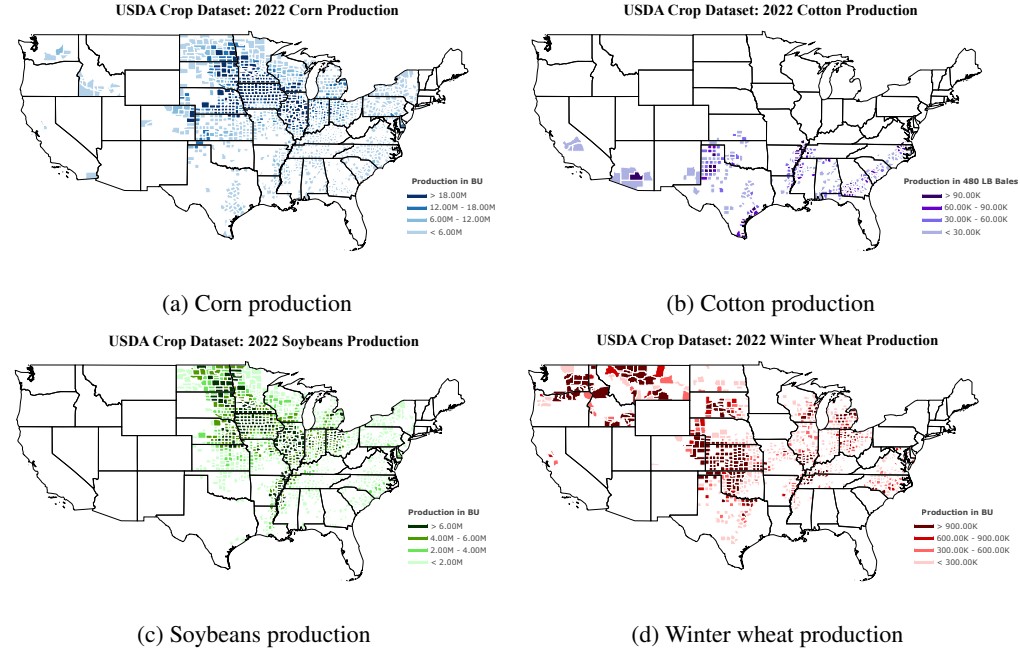

Figure 13: Illustration of USDA Crop Dataset for (a) 2022 corn production, (b) 2022 cotton production, (c) 2022 soybeans production, and (d) 2022 winter wheat production.

## C   DETAILS OF DATA COLLECTION

### C.1   SIGNIFICANCE OF OUR CLOUD COVERAGE SETTING AND REVISIT FREQUENCY FOR SENTINEL-2 IMAGERY

This section supplementss Section 3.3 of the main paper by demonstrating the necessity and importance of our cloud coverage setting (*i.e.*, $\leq 20\%$) and revisit frequency (*i.e.*, 14 days) for Sentinel-2 Imagery. Figures 14 and 15 present examples of Sentinel-2 Imagery under the original revisit fre-

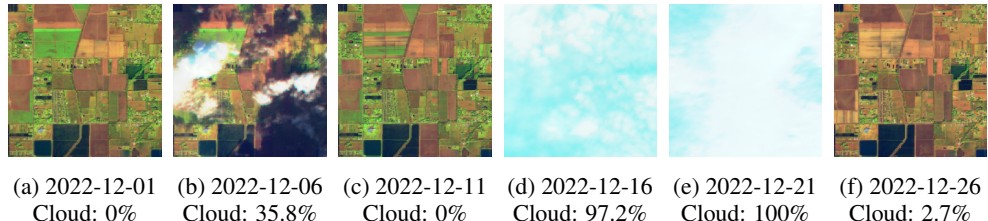

| (a) 2022-12-01 | (b) 2022-12-06 | (c) 2022-12-11 | (d) 2022-12-16 | (e) 2022-12-21 | (f) 2022-12-26 |
| Cloud: 0% | Cloud: 35.8% | Cloud: 0% | Cloud: 97.2% | Cloud: 100% | Cloud: 2.7% |

Figure 14: Examples of Sentinel-2 Imagery under the original revisit frequency of 5 days without our cloud coverage setting, with the revisit date and the cloud coverage listed below each image.

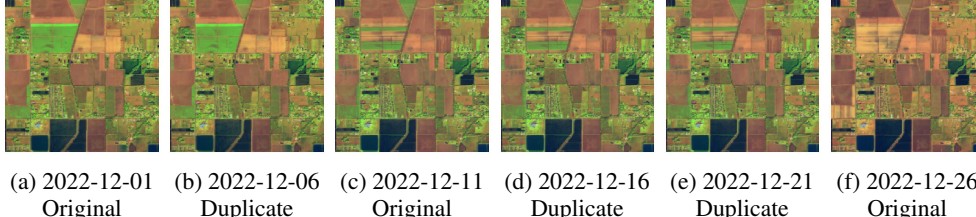

| (a) 2022-12-01 | (b) 2022-12-06 | (c) 2022-12-11 | (d) 2022-12-16 | (e) 2022-12-21 | (f) 2022-12-26 |
| Original | Duplicate | Original | Duplicate | Duplicate | Original |

Figure 15: Examples of Sentinel-2 Imagery under the original revisit frequency of 5 days and our cloud coverage setting. The revisit date is listed below each image. "Duplicate" (or "Original") indicates whether the satellite image is duplicate (or not) under our cloud coverage setting.

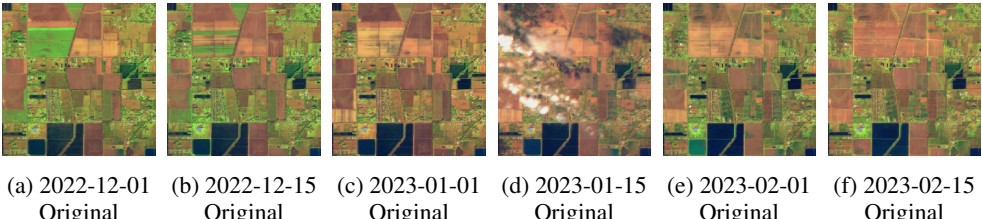

| (a) 2022-12-01 | (b) 2022-12-15 | (c) 2023-01-01 | (d) 2023-01-15 | (e) 2023-02-01 | (f) 2023-02-15 |
| Original | Original | Original | Original | Original | Original |

Figure 16: Examples of Sentinel-2 Imagery under our revisit frequency of 14 days and our cloud coverage setting, with the revisit date listed below each image. We would like to highlight that there are no duplicate satellite images observed.

quency of 5 days with and without our cloud coverage setting, respectively. Figure 16 illustrates satellites images under our revisit frequency of 14 days and our cloud coverage setting (*i.e.*, $\leq 20\%$).

From Figure 14, we observed that the cloud coverage may significantly impair the quality of Sentinel-2 Imagery (see Figures 14b, 14d, and 14e). Worse still, the extreme cases of cloud coverage (refer to Figures 14d and 14e) degrade satellite images into noisy representations. This demonstrates the significance of our cloud coverage setting for discarding low-quality satellite images. Unfortunately, under the original sentinel-2 revisit frequency of 5 days, our cloud coverage setting would result in a large proportion of duplicate satellite images, *e.g.*, 50% (*i.e.*, 3 out of 6 satellite images) as depicted in Figure 15 [3]. This is because if the cloud coverage in our requested revisit day exceeds 20%, Processing API Sentinel-Hub (2023) will download the most recent available satellite images, whose cloud coverage satisfies our condition (*i.e.*, $\leq 20\%$). In sharp contrast, extending the revisit frequency from 5 days to 14 days markedly decreases the occurrence of duplicate satellite images. For example, there are no duplicate satellite images observed in Figure 16. Hence, our revisit frequency of 14 days for Sentinel-2 Imagery is necessary as it can significantly improve storage and training efficiency.

## C.2   COUNTY PARTITIONING

In Section 3.3 of our main paper body, we have introduced partitioning one county into multiple high-spatial-resolution grids for precise agricultural tracking. Here, we provide the details for such a partition. A naive way to achieve this is to expand a county's geographic boundary to a rectangle area by using its maximal and minimal latitude and longitude, and then evenly divide such a rectangle

---

[3]Figures 15a and 15b depict identical satellite images. Likewise, Figures 15c, 15d, and 15e also display the same satellite images.

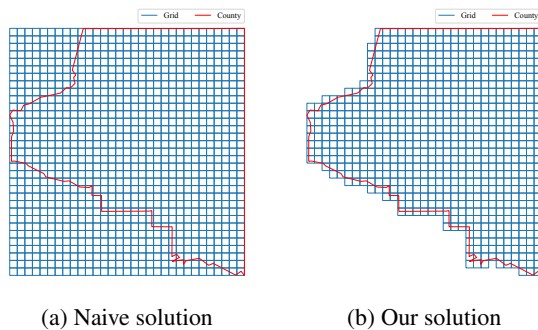

(a) Naive solution          (b) Our solution

Figure 17: Difference between the naive solution and our solution. (a) The naive solution leads to a significant number of grids falling outside the county's polygon. (b) By using our solution, the boundaries of grids (*i.e.*, the blue line) align perfectly with the county's boundary (*i.e.*, the red line).

Table 4: Details of WRF-HRRR Computed Dataset

| Source | Parameters | Description |
|---|---|---|
| WRF-HRRR model | Averaged Temperature | 2 metre averaged temperature during a day/month. Unit: K |
| | Precipitation | Total precipitation. Unit: $\mathrm{kg/m^2}$ |
| | Relative Humidity | 2 metre relative humidity. Unit: % |
| | Wind Gust | Wind gust on the ground. Unit: $\mathrm{m/s}$ |
| | Wind Speed | Wind speed on the ground. Unit: $\mathrm{m/s}$ |
| | Downward Shortwave Radiation Flux | The total amount of shortwave radiation that reaches the Earth's surface. Unit: $\mathrm{W/m^2}$ |
| Computed by us | Maximal Temperature | 2 metre maximal temperature during a day/month. Unit: K |
| | Minimal Temperature | 2 metre minimal temperature during a day/month. Unit: K |
| | Vapor Pressure Deficit (VPD) | The amount of drying power the air has upon the plant. Unit: kPa |

area into multiple grids. Unfortunately, such a partition solution may result in a large number of grids outside the county polygon for some large counties (see Figure 17a). To handle this matter, we develop a novel solution by dropping the grids outside the county's boundary (see Figure 17b). Compared to the naive solution, our solution enjoys two advantages. First, it can significantly reduce the disk space storage size. Take Coconino County in Arizona for example, by employing our solution, its total number of grids degrades from 1023 to 729, which is 0.71x less than that from the naive solution. Second, our solution can evade the negative effect incurred by regions outside the county's boundary on crop yield predictions.

## C.3 DETAILS OF WRF-HRRR COMPUTED DATASET

Table 4 presents details of meteorological parameters in the WRF-HRRR Computed Dataset, where 6 weather parameters, *i.e.*, averaged temperature, precipitation, relative humidity, wind gust, wind speed, downward shortwave radiation flux, are obtained directly from the WRF-HRRR model, and 3 other parameters, *i.e.*, maximal temperature, minimal temperature, vapor pressure deficit (VPD), are calculated by ourselves. Notably, VPD describes the difference between the amount of moisture in the air and the maximum amount of moisture the air can hold at a specific temperature, which is an important concept in understanding the environmental conditions that affect plant growth and transpiration. Given two meteorological parameters, *i.e.*, the temperature measured in Kelvin $T_K$ and the relative humidity $RH$, VPD is calculated by the following equations:

$$
\begin{aligned}
T_C &= T_K - 273.15, \\
VP_{\mathrm{sat}} &= \frac{610.7 \times 10^{(7.5 \times T_C)/(237.3 + T_C)}}{1000}, \\
VP_{\mathrm{air}} &= VP_{\mathrm{sat}} \times \frac{RH}{100}, \\
VPD &= VP_{\mathrm{sat}} - VP_{\mathrm{air}}.
\end{aligned} \tag{1}
$$

C.4    SPATIAL AND TEMPORAL ALIGNMENT OF OUR CROPNET DATASET

Here, we present an example of our JSON configuration file (see Listing 4) for one U.S. county (*i.e.*, Baldwin in Alabama), to show how satellite images from Sentinel-2 Imagery, daily and monthly weather parameters from the WRF-HRRR Computed Dataset, and the crop information from USDA Crop Dataset, are spatially and temporally aligned. As presented in Listing 4, "data.sentinel" and "data.HRRR.short_term" respectively represent satellite images and daily meteorological parameters during the crop growing season, "data.HRRR.long_term" indicates monthly weather conditions from previous 5 years, and "data.USDA" provides the crop information for the county. Meanwhile, "FIPS" and "year" respectively indicate the unique FIPS code and the year for the growing season, enabling us to obtain the data for our targeted county in a specific year. In summary, the JSON configuration file allows us to retrieve all three modalities of data over the time and region of interest.

```json
{
    "FIPS":"01003",
    "year":2022,
    "county":"BALDWIN",
    "state":"AL",
    "county_ansi":"003",
    "state_ansi":"01",
    "data":{
        "HRRR":{
            "short_term":[
                "HRRR/data/2022/AL/HRRR_01_AL_2022-04.csv",
                "HRRR/data/2022/AL/HRRR_01_AL_2022-05.csv",
                "HRRR/data/2022/AL/HRRR_01_AL_2022-06.csv",
                "HRRR/data/2022/AL/HRRR_01_AL_2022-07.csv",
                "HRRR/data/2022/AL/HRRR_01_AL_2022-08.csv",
                "HRRR/data/2022/AL/HRRR_01_AL_2022-09.csv"
            ],
            "long_term":[
                [
                    "HRRR/data/2021/AL/HRRR_01_AL_2021-01.csv",
                    "HRRR/data/2021/AL/HRRR_01_AL_2021-02.csv",
                    "HRRR/data/2021/AL/HRRR_01_AL_2021-03.csv",
                    "HRRR/data/2021/AL/HRRR_01_AL_2021-04.csv",
                    "HRRR/data/2021/AL/HRRR_01_AL_2021-05.csv",
                    "HRRR/data/2021/AL/HRRR_01_AL_2021-06.csv",
                    "HRRR/data/2021/AL/HRRR_01_AL_2021-07.csv",
                    "HRRR/data/2021/AL/HRRR_01_AL_2021-08.csv",
                    "HRRR/data/2021/AL/HRRR_01_AL_2021-09.csv",
                    "HRRR/data/2021/AL/HRRR_01_AL_2021-10.csv",
                    "HRRR/data/2021/AL/HRRR_01_AL_2021-11.csv",
                    "HRRR/data/2021/AL/HRRR_01_AL_2021-12.csv"
                ],
                # The remaining years are hidden for conserving space
                ...
            ]
        },
        "USDA":"USDA/data/Soybeans/2022/USDA_Soybeans_County_2022.csv",
        "sentinel":[
            "Sentinel/data/AG/2022/AL/Agriculture_01_AL_2022-04-01_2022
-06-30.h5",
            "Sentinel/data/AG/2022/AL/Agriculture_01_AL_2022-07-01_2022
-09-30.h5"
        ]
    }
}
```

Listing 4: Example of our JSON configuration file.

Table 5: Ablation studies for different modalities of the CropNet dataset, with five scenarios considered and the last row presenting the results by using all modalities

| Modality | Scenario | Corn | | | Soybeans | | |
|---|---|---|---|---|---|---|---|
| | | RMSE ($\downarrow$) | $R^2$ ($\uparrow$) | Corr ($\uparrow$) | RMSE ($\downarrow$) | $R^2$ ($\uparrow$) | Corr ($\uparrow$) |
| Sentinel-2 Imagery | w/o temporal images | 22.1 | 0.758 | 0.870 | 5.72 | 0.773 | 0.879 |
| | w/o high-resolution images | 27.9 | 0.656 | 0.810 | 7.80 | 0.631 | 0.794 |
| WRF-HRRR Computed Dataset | w/o WRF-HRRR data | 20.6 | 0.758 | 0.871 | 5.78 | 0.764 | 0.874 |
| | w/o short-term data | 18.6 | 0.796 | 0.892 | 5.04 | 0.816 | 0.903 |
| | w/o long-term data | 15.3 | 0.854 | 0.924 | 4.72 | 0.825 | 0.908 |
| All | — | 13.2 | 0.890 | 0.943 | 3.91 | 0.879 | 0.937 |

# D  SUPPORTING EXPERIMENTAL SETTINGS AND RESULTS

## D.1  ADDITIONAL EXPERIMENTAL SETUPS

**CropNet Data.** Due to the limited computational resources, we are unable to conduct experiments across the entire United States. Consequently, we extract the data with respect to five U.S. states, *i.e.*, Illinois (IL), Iowa (IA), Louisiana (LA), Mississippi (MS), and New York (NY), to exhibit the applicability of our crafted CropNet dataset for county-level crop yield predictions. Specifically, two of these states (*i.e.*, IA and IL) serve as representatives of the Midwest region, two others (*i.e.*, LA and MS) represent the Southeastern region, and the fifth state (*i.e.*, NY) represents the Northeastern area. Four of the most popular crops are studied in this work, *i.e.*, corn, cotton, soybeans, and winter wheat. For each crop, we take the aligned Sentinel-2 Imagery and the daily data in the WRF-HRRR Computed Dataset during growing seasons in our CropNet dataset, respectively for precise agricultural tracking and for capturing the impact of growing season weather variations on crop growth. Meanwhile, the monthly meteorological parameters from the previous 5 years are utilized for monitoring and quantifying the influence of climate change on crop yields.

## D.2  SIGNIFICANCE OF EACH MODALITY OF OUR CROPNET DATASET

To show the necessity and significance of each modality data in our CropNet dataset, we examine five scenarios. First, we drop the temporal satellite images (denoted as "w/o temporal images") by randomly selecting only one day's imagery data. Second, we discard the high-resolution satellite image (denoted as "w/o high-resolution images") by using only one satellite image to capture the whole county's agricultural information. Third, we ignore the effects of weather variations on crop yields by dropping all meteorological data, denoted as "w/o WRF-HRRR data". Similarly, "w/o short-term data" and "w/o long-term data" represent masking out the daily and monthly meteorological parameters, respectively. We also include prediction results by using all modalities of the CropNet (denoted as "All") for performance comparison. Note that the USDA Crop Dataset provides the label for crop yield predictions; hence, no ablation study requires.

Table 5 presents the experimental results under the MMST-ViT model Lin et al. (2023). We have four observations. First, discarding the temporal satellite images (*i.e.*,"w/o temporal images") degrades performance significantly, raising the RMSE value by 8.9 (or 1.81) and lowering the Corr value by 0.073 (or 0.058) for corn (or soybeans) yield predictions. This is due to that a sequence of satellite images spanning the whole growing season are essential for tracking crop growth. Second, "w/o high-resolution images" achieves the worst prediction performance, with a largest RMSE vaue of 27.9 (or 7.8) and a lowest Corr value of 0.810 (or 0.794) for corn (or soybeans) yield predictions. The reason is that high-resolution satellite images are critical for precise agricultural tracking. Third, dropping meteorological parameters (*i.e.*, w/o WRF-HRRR data) makes MMST-ViT fail to capture meteorological effects on crop yields, leading to the increase of RMSE value by 7.4 (or 1.87) and the decease of Corr value by 0.072 (or 0.063) for predicting corn (or soybeans) yields. Fourth, discarding either daily weather parameters (*i.e.*,"w/o short-term data") or monthly meteorological parameters (*i.e.*,"w/o long-term data") lowers crop yield prediction performance. The reason is that the former is necessary for capturing growing season weather variations, while the latter is essential for monitoring long-term climate change effects. Hence, we conclude that each modality in our CropNet dataset is important and necessary for accurate crop yield predictions, especially for those crops which are sensitive to growing season weather variations and climate change.

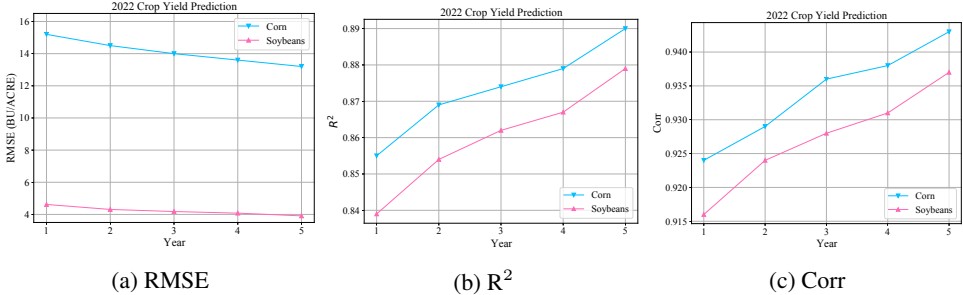

(a) RMSE        (b) R$^2$        (c) Corr

Figure 18: Performance of 2022 corn and soybeans yield predictions by varying the length of long-term meteorological data from one year to five years.

### D.3 IMPACT OF LONG-TERM METEOROLOGICAL DATA

We further quantify the effects of long-term meteorological data by varying its length from one year to five years, for supporting Section 4 of the main paper. Figure 18 depicts the performance results for 2022 corn (*i.e.*, the blue line) and soybeans (*i.e.*, the pink line) yield predictions under the MMST-ViT model. We observed that lengthening the monthly meteorological data from one year to five years can improve the performance of crop yield predictions, lowering the RMSE value by 2.2 (or 0.7) and lifting R$^2$ and Corr values respectively by 0.035 (or 0.040) and by 0.019 (or 0.021), for corn (or soybeans) yield predictions. The reason is that a longer range of monthly meteorological parameters can better monitor and relate the climate change effect on crop growth.

