# OpenReview forum: "CropNet: An Open Large-Scale Dataset with Multiple Modalities for Climate Change-aware Crop Yield Predictions"
_ICLR.cc/2024/Conference — Submitted to ICLR 2024_

### Official Review · Reviewer_m3sL · 2023-10-25

**Soundness:** 1 poor
**Presentation:** 1 poor
**Contribution:** 2 fair
**Rating:** 3
**Confidence:** 4

**Summary:**

This work presents a new large scale Crop Yield Prediction dataset, that not only includes Sentinel-2 satellite imagery but also meteorological data with target crop yield values from the USDA. In contrast to other listed previous works this offers the advantage of additional context data that could enhance crop yield predictions. The dataset is additionally accompanied by a PyPi package to facilitate an easier user experience for practitioners in this field.

**Strengths:**

The CropNet dataset creation shows a substantial and noteworthy effort that is a real contribution to the wider Machine Learning community. The description of how the authors curated the dataset itself is mostly clear, although there are some missing pieces of information. Figures 2-5 are very helpful and convey the dataset well.

**Weaknesses:**

While the CropNet dataset itself appears to be a good contribution, there should be a more elaborate discussion about differences to existing datasets in this realm. Additionally, I see several issues with the experiment setting and the conclusions drawn from it.

Dataset Discussion:
Your main focus and argument for this dataset is the fact, that it is combines the three data modalities (whereas other works only have a subset of those available) and that it is much larger in terms of storage size in GBs. However, as a "dataset paper" a discussion and comparison around the context in which this dataset is placed needs to be more elaborate in my opinion.
- Potentially missing references to these datasets [Iizumi and Sakai 2020](https://www.nature.com/articles/s41597-020-0433-7) and [EarthStat](http://www.earthstat.org/)
- while storage size can be a feature of comparing datasets, it is also somewhat arbitrary since it depends on many factors, like the file format, numerical datatype etc. So more GB is not simply better as you sometimes stress with the "terrabytes-sized" phrase. For example, if your stored satellite imagery would contain some additional S2 band, its memory footprint would be larger, but would it better for crop yield prediction? Vice versa, if there was a more memory efficient way to store your dataset, would it be less valuable?
- More important and necessary is a more extensive discussion about other features of the dataset and how that falls in the context of existing works. One factor would be spatial coverage, which is a tremendous issue in the geospatial ML world, where the vast majority of datasets are curated based on datasets in the Western World, but are lacking in areas where they might actually have the largest impact, Other factors, such as temporal coverage, the number of crops included, etc. should have a more central focus in your discussion

Experiments:
The experiment section is where I see the most issues and is the dominant factor behind my review rating. To begin with many important details in your evaluation and analysis are missing or inaccurate:
- how do you split your data in train, validation, and test set? Multiple ways exist, but their strengths and weaknesses are of great importance in spatio-temporal data to make statements about model generalization
- How do you obtain error bars in Figure 7 for the results? Are these computed over repeated model runs for example? But then they are missing in Table 3, which does not have any interval indication.
- In section 3.2 you state as your first observation that "First, all models achieve superb prediction performance with our CropNet data". My issue with this statement is two-fold: One, "superb" compared to what? What are existing approaches that are outperformed by these methods. More generally, how is crop yield prediction done at the USDA at the moment to which you could compare your approach. Second, you make it seem like the "superb prediction performance" you are seeing is due to the CropNet dataset, but this is misleading. If you train models on a dataset and see great metrics, that does not mean your dataset is "great". Quantitative evaluation of models and the quality of the dataset in this context should be discussed separately. The quality of the dataset has of course impact to what a ML model can learn, however, just looking at model metrics do not prove that you have a great dataset.
- As your second observation in 3.2 you state that the MMST-ViT performance is due to the transformer attention mechanism, but this is merely speculative. You have no explicit experiment to underline this claim. This is of course not trivial to do, but you should not state it like a fact. You make a similar claim under 3.3 for MMST-ViT's performance (last paragraph). You make another claim like that in 3.4 last paragraph, where you do not show prove, but are speculating.
- In section 3.4 and Figure 8 you seem to argue that SSL helps with overfitting, however, you are giving no experimental support for that conclusion. Overfitting is a phenomena between a model's performance on the training data and some held out data, a validation dataset, where model performance on the training data is "better" than on the validation dataset. Figure 8 or the text does not mention which dataset split the reported values stem from. This is problematic because you seem to suggest that because MM-SSL achieves lower metric values, you have reduced overfitting, but there is no reference to what that "overfitting" gap between training and validation set is. Therefore, section 3.4 is extremely misleading given the current level of detail or even wrong if you suggest that just a lower metric on an evaluation set reduces overfitting
- I am also not sure why you motivate SSL through the lense of overfitting. In Geospatial ML the possibly greatest advantage of a SSL pretraind crop yield model would be its ability to generalize well over other areas where you have little data available but would like to make predictions. Here, models trained on just little data might be prone to overfitting, but that is different from your evaluation you are describing in your paper.

**Questions:**

Some terminology is confusing or misleading, for example:
- Sentinel-2 imagery does not provide categories of images. It provides multi spectral data, which offers different analysis opportunities. The NDVI is something that is computed from two multi spectral bands (B08 and B04) but it is not a category of images provided. Similarly "agriculture imagery" is not a Sentinel-2 category.
- Seciton 2.2, page 4 and Table 2: Spatial Resolution of Sentinel 2 imagery usually refers to pixel level resolution, which for sentinal-2 is 10m, 20m, or 60m, maybe your county single grid box has that 9x9 km resolution, but sentinel-2 imagery certainly does not. Are you meaning to say something like spatial extent?

Methodological questions:
- How do you deal with no data areas that appear in satellite imagery that can appear due to their swath?
- Leaving the Deep Learning world, how are crop yield predictions currently being made at a county level, what are the metrics they use, what are the data sources and models, how would they compare? I would like to know what the current approach of institutions for crop yield prediction is so that there is an indication of how Machine Learning could help
- why are you selecting the three spectral bands B02, B08, B11 as your "agricultural imagery"

Experiment/Results questions:
- How does taking the square root of RMSE enhance the visualization? It makes the values smaller, but you refer to them in the text as if they were actual RMSE values and it is confusing given that RMSE values are mentioned in multiple places
- In section 3.3 you mention "averaged RMSE values". But I am confused about "averaged" over what? RMSE is already a single value describing a quantitative performance metric over all data points in an evaluation set given its mathematical definition.

---

### Official Review · Reviewer_ME5N · 2023-10-29

**Soundness:** 1 poor
**Presentation:** 1 poor
**Contribution:** 1 poor
**Rating:** 1
**Confidence:** 5

**Summary:**

The paper proposes a large-scale dataset for crop yield prediction from Sentinel-2, Meteorological input data to predict crop yield at a County over the USA. No methodological questions are investigated and the insights are limited. Even though "climate change aware" crop yield is mentioned in the title, no measures towards efficiently integrating these meteorological idata into the tested deep learning models is discussed. While this paper may be a suitable candidate to the NeurIPS data track, its value to ICLR is unclear. Still, even as pure dataset paper, it has several limitations and unclarities that need to be outlined more clearly to highlight its important, even for the domain field of crop yield prediction:

1. **dataset size**: the paper sees its size in the terrabytes as an advantage. However, it is unclear what benefits this high quantity of data would contribute without further investigation. Collecting large datasets from public sources is not difficult, but creating a high-quality dataset is. The quality of this dataset is not discussed throughout the paper.
2. **crop-yield data**: the USDA crop yield data is at county scale. Yield data at this coarse aggregation available since years. Similarly in Europe, Eurostats publishes NUTS-3 "county-level" crop yield data. Crucually, yield data at this level is not particularly helpful, while the more relevant and challenging topic of field-level crop yield estimation, is not discussed in this paper at all.
3. **satellite data**: the paper states that it uses Sentinel-2 imagery, which has a resolution of 10-60m. This data seems to be massively downsampled to a 9 km by 9 km resolution. Why did the authors not use a more suitable sensor at this low resolution, like MODIS (1km) resolution that is much better suited for this kind of county-level yield prediction? Also, the paper states that it uses "AG" images. However, this seems to be a SWIR-NIR-Blue composite. What makes this false-color composite more suited for agriculture and why were not more Sentinel-2 bands used?
4. **experiments** even though this dataset was designed with "deep-learning friendliness" in mind, the details of input-data and output dimensions is not discussed. For instance, What is the dimension of the input tensor? How are the (much lower resolution) meteorological data integrated in this input tensor?
5. **"deep learning friendlyness"**: the paper argues that these available datasets are not "deep-learning friendly". However, it is not clear in what way this dataset more applicable for deep learning applications aside from providing a python and pytorch wrapper around the data.
4. **references and related work**: the discussion of related work is quite limited. First, the paper mixes crop type classification (PASTIS; DENETHOR) datasets with its crop yield estimation objective? Is it clear that crop type mapping and crop yield estimation are different research fields with different inherent challenges? Similarly, field-wise crop yield estimation is done in the US throught the Scalable Crop Yield Mapper (SCYM) (Lobell et al., 2015), which is done in Google Earth Engine, and does not require downloading terrabytes of data. It is surprising that this (and related works) are not referenced at all in this paper even though they share exactly the same area of interest. Deep learning approaches by, for instance, You et al (2017) are also not referenced at all.
6. **preprocessing** the paper frames several standard Geo-Information System (GIS) processing algorithms as innovations (e.g., see Fig 17 "Naive solution" vs "Our approach"). These can be implemented in geopandas or postgis in normal data loaders and are usually not stated explicitly in the papers.
7. **accessibility**, it is not clear if the "API" provided in the package is served from a server infrastructur, such as Google Earth Engine, or Microsoft Planetary Computer, of if the users have to download the entire datasets first locally. If this is the case, what makes this API different to a standard dataloader class?

Lobell, D. B., Thau, D., Seifert, C., Engle, E., & Little, B. (2015). A scalable satellite-based crop yield mapper. Remote Sensing of Environment, 164, 324-333.

You, J., Li, X., Low, M., Lobell, D., & Ermon, S. (2017, February). Deep gaussian process for crop yield prediction based on remote sensing data. In Proceedings of the AAAI conference on artificial intelligence (Vol. 31, No. 1).

**Strengths:**

* crop yield prediction is an important topic that should be investigated more with modern approaches.

**Weaknesses:**

* pure dataset contribution with little methodological components or insights
* also the dataset seems to be focused on quantity consideration, while lacking detail and quality. It is not clear if this dataset was created with domain experts from the fields of remote sensing or crop yield prediction, as design choices are not well-supported (like Sentinel-2 over MODIS) and central papers/references from crop yield prediction (e.g., SCYM) are not mentioned.

**Questions:**

* what are the methodological insights that this dataset can provide? Towards what machine learning problems, is this dataset taylored towards?
* what are the dimensions of the data inputted to the deep learning models for the experiments? In particular, how were remote sensing and meteorological data merged?
* why was Sentinel-2 chosen over MODIS, which as a native resolution much closer to the 9 by 9 km resolution of the dataset?

---

### Official Review · Reviewer_yC3M · 2023-11-01

**Soundness:** 3 good
**Presentation:** 2 fair
**Contribution:** 3 good
**Rating:** 6
**Confidence:** 3

**Summary:**

The paper highlights the significant gap in existing AI research for accurate crop yield predictions due to the absence of comprehensive, multi-modal datasets. To address this, the authors present the CropNet dataset, a pioneering terabyte-sized, multi-modal resource tailored for crop yield predictions across the contiguous U.S. at the county level, integrating Sentinel-2 Imagery, WRF-HRRR Computed Dataset, and USDA Crop Dataset from 2017-2022. Preliminary experiments using deep learning on CropNet affirm its potential for effective, climate-sensitive crop yield forecasting.

**Strengths:**

1. A very rich dataset, compared to other datasets related to this task, specially in terms of modality.
2. Good writeup style, in terms of grammar and readability.
3. This work has good reproducibility and impact towards humanity and also deep learning and agriculture communities.

**Weaknesses:**

1. There is no discussion section.
2. There is no limitation mentioned in the paper.
3. Not enough literature review.

**Questions:**

1. Is Section 2.1 needed, since it is mentioned more or less in Section 1? I would suggest adding a few works related to yours and discussing how your work overcomes those.
2. Section 3: You should add the full form of CNN, RNN, GNN, MMST, MM-SSL, ViT before using such a short form.
3. Please add the limitation subsection.
4. Please add the discussion section.  I would highly suggest adding it.
5. Table 3: It is better to bold the best values.
6. Instead of showing the temperature as K, it is better to replace it with either Celcius or Farhenheit.

---

### Official Review · Reviewer_ma3c · 2023-11-03

**Soundness:** 2 fair
**Presentation:** 2 fair
**Contribution:** 2 fair
**Rating:** 3
**Confidence:** 5

**Summary:**

This papers presents a dataset for crop yield prediction, consisting of 3 parts: satellite imagery, weather data and crop yield data.

the authors provide evaluation of the dataset with the state-of-the art DL models (ConvLSTM, CNN-RNN, GNN_RNN and MMST-ViT). The performance of the models evaluated with 3 different metrics.

**Strengths:**

This work represents a useful tool for machine learning researchers, including satellite, weather and ground observation data.
The models that are used for evaluation are state-of-the art methods, which predict crop yields in 2 modalities: for 2022 and one-year-ahead.

**Weaknesses:**

The proposed dataset uses a few assumptions, which might be restrictive for dataset usage:
1) they provide only 4 of 12 bands of Sentinels-2 imagery (the motivation for this assumption is not provided)
2) only Sentinel -2 L1-C processing level is provided (although some researchers might find L2-A processing level more useful)
3) the resolution of Sentinel-2 imagery is decreased from 10x10 meters per pixel to 9x9 km per pixel (the motivation for this assumption is not clearly explained)
4) the comparison of the deep learning approaches with the conventional ,methods is not provided (in case of the yield prediction sometimes models like RF perform better)
5) the accuracy of the provided predictions does not outperform existing methods
6) the novelty of the dataset is not justified, from the provided description the dataset is a combinations of the existing dataset with some automation and reduction in functionality

**Questions:**

1) why do you choose the AG and NDVI indices only? some research shows that usage of all 12 bands provides better information for crop yield prediction
2) how does your predictions compare with traditional data assimilation methods?
3) please refrain the from the phrase "somewhat mediocre prediction performance" and provide the quantitative comparison of the performance of the proposed methods/datasets with the previous work
4) explanation of why the cloud coverage decreases the model performance is unnecessary, please provide the description of the cloud masking method, used in the paper instead

---

### Meta-Review · Area_Chair_2ydz · 2023-12-09

**Metareview:**

The author present a multi-modal crop yield prediction dataset, CropNet. Which this topic is very timely and impactful, the reviewers feel there to be a number of critical weaknesses, and the authors have not offered a rebuttal. Therefore, I recommend rejection.

**Justification For Why Not Higher Score:**

The reviewers are largely agreed on rejection.

**Justification For Why Not Lower Score:**

n/a

---

### Decision · Program_Chairs · 2024-01-16

Reject